# Recent Advances in Ferroelectric Materials-Based Photoelectrochemical Reaction

**DOI:** 10.3390/nano12173026

**Published:** 2022-08-31

**Authors:** Limin Yu, Lijing Wang, Yanmeng Dou, Yongya Zhang, Pan Li, Jieqiong Li, Wei Wei

**Affiliations:** 1Henan Engineering Center of New Energy Battery Materials, Henan D&A Engineering Center of Advanced Battery Materials, Shangqiu Normal University, Shangqiu 476000, China; 2Shandong Yuhuang New Energy Technology Co., Ltd., Heze 274000, China

**Keywords:** ferroelectric, polarization, photoelectrochemical, recent advances

## Abstract

Inorganic perovskite ferroelectric-based nanomaterials as sustainable new energy materials, due to their intrinsic ferroelectricity and environmental compatibility, are intended to play a crucial role in photoelectrochemical field as major functional materials. Because of versatile physical properties and excellent optoelectronic properties, ferroelectric-based nanomaterials attract much attention in the field of photocatalysis, photoelectrochemical water splitting and photovoltaic. The aim of this review is to cover the recent advances by stating the different kinds of ferroelectrics separately in the photoelectrochemical field as well as discussing how ferroelectric polarization will impact functioning of photo-induced carrier separation and transportation in the interface of the compounded semiconductors. In addition, the future prospects of ferroelectric-based nanomaterials are also discussed.

## 1. Introduction

Development and utilization of clean, renewable solar energy has proven to be an efficient way to cope with environmental concerns. Photocatalysis [1,2], photoelectrochemical (PEC) water splitting [3,4] and photovoltaic [5] are topics of effective use of solar energy with the central step of PEC. The photoactive species were categorized as inorganic semiconductors [6,7,8,9], organic semiconductors [10,11], organic-inorganic hybrid materials [12,13], polymer semiconductor materials [14], perovskites [15,16,17] and so on. Perovskite oxide ferroelectrics, generally referring to ABO_3_ and its derivatives, possess intrinsic ferroelectric polarization (*P*) with reversible external electric fields that have come to the forefront of nanoscience and nanotechnology ever since the pioneering discovery [18,19], followed by other ferroelectric ceramics, such as KNbO_3_, KTaO_3_, PbTiO_3_, etc. The ferroelectricity is due to the asymmetry of the crystal structure with the existence of electric dipoles inside because of the misalignment of the positive and negative charge centers [20]. Therefore, ferroelectrics’ splendid photoelectric properties include large nonlinear optical coefficients, stable chemical properties, as well as adjustable band gap. Furthermore, because of the excellent separation and migration efficiency of electron-hole pairs under depolarization field, photo-generated voltage above the energy gap can be obtained [21]. Despite the above-mentioned advantages, there are still some problems need to be investigated. For example, the low transport efficiency of the optical carrier has led to the knockdown quantum efficiency (10^−5^), which is far lower than conventional P-N junction devices (10^−1^). This phenomenon is mainly caused by two factors: (1) ferroelectrics can only absorb ultraviolet light because of wide bandgap values. (2) The low electrical conductivity properties seriously hinder the diversion of photo-generated electron holes, whereas the increment of specific conductance will cause electrical leakage inside the material so as that it cannot maintain strong electrical polarization [22]. Therefore, ion doping is adopted to reduce the bandgap. The combination of semiconductor and ferroelectrics is an efficient method to improve the separation efficiency of electron holes and will reduce the need for interface state control because interface is not a necessary channel of photo-carriers [23]. In all, as evidenced by gradual increasing academic papers, the advanced ferroelectric materials for photo-electric conversion are still currently in progress.

Given the pace of advances in this field, the ferroelectric materials for photovoltaic reactions and photocatalysis are the subject of the present review. Specifically, this work surveys the functional materials in a period of rapid development of ABO_3_ and 2D ferroelectrics. The recent progress, research technique, and future prospects of this orientation are also discussed and evaluated. We sincerely apologize to the researchers of ferroelectric-based photovoltaic and photocatalysis publications that were not deliberately overlooked and want to remind our readers that this article is not designed to cover comprehensive theoretical knowledge of ferroelectric-based photovoltaic reactions and photocatalysis but rather to present several typical ferroelectric materials. Furthermore, we do our utmost to emphasize some important publications which can accelerate further research in depth in this area. For learners seeking to acquire more knowledge on the fundamental theory about photovoltaic reactions and photocatalysis, we recommend they seek more information in other reviews [24,25,26,27,28].

## 2. Recent Advances of Different Species

The structure of ferroelectrics for photovoltaic reactions and photocatalysis comes in two kinds, with one category of perovskite-structured materials and another layered crystal structure. Generally, perovskite-structured materials are highly symmetrical in structure with a typical chemical formula ABO_3_, in which the cube’s corner points (A-site) was occupied by univalent or divalent metal ions, while the center of the cube (B-site) was taken up by metal ions and oxygen ions occupying the face-center position of the cube. Layered structure materials refer to 2D materials and often consist of either a single or a few atomic layers and several representative ferroelectric materials are listed below.

### 2.1. BaTiO_3_

BaTiO_3_, with the bandgap of 3.18 eV, is a widely used ferroelectric material. Steve Dunn proved that BaTiO_3_ with tetragonal crystal reveals a 3-fold increase in the Rhodamine B degradation compared with cubic material [29]. Although pure BaTiO_3_ owns a certain photocatalytic property, the wide band-gap precludes its photo-activity. Several strategies were applied for optimizing its optoelectronic properties: (1) loading noble metals onto the surface of ferroelectrics. Li et al. [30] assembled Ag loaded BaTiO_3_ nanotube arrays system and Ag act as the photogenerated electron traps, can inhibits the recombination of photoelectron and holes, leading to the increment of photo-degradation of MO efficiency. (2) doping ion, especially transition metal ions, can effectively increase the lifetime of carriers, reduce the recombination rate of electron hole pairs and increase the degree of separation of electron hole pairs. A certain number of studies reported that ion-doping, such as Li^+^ [31], Fe^3+^ [32], Mn^2+^ [33], Y^3+^ [34] and even NaNbO_3_ [35], KNbO_3_ [36], affect the crystal structure of intrinsic materials. (3) combining two semiconductors with apposite energy level and bandgap can change band bending, widen the light absorption range and accelerate carrier separation. As shown in Figure 1a,b, Wang et al. [37] reported that TiO_2_/BaTiO_3_ core/shell NWs obtained 67% photocurrent enhancement compared with pure TiO_2_ NWs, which can ascribe to adjustable ferroelectric polarization by external electric field poling, and this research proved that ferroelectric or piezoelectric potential-induced band structure engineering likely raise the capability of PEC electrode. Hu et al. [38] synthesized necklace-like BaTiO_3_ nanofibers by sol-gel assisted electrospinning and subsequently coating TiO_2_ on the nanofiber surface by wet-chemical. The TEM of necklace-like BaTiO_3_ nanofibers was shown in Figure 1c, which exhibited both high piezoelectric coefficient and ferroelectric. Due to the excellent piezo-photocatalytic property, the necklace-like BaTiO_3_@TiO_2_ core-shell demonstrated outstanding degradation of MO under both UV irradiation and ultrasound, as shown in Figure 1d.

Besides TiO_2_, other oxide semiconductors are also adopted to combine with BaTiO_3_, such as α-Fe_2_O_3_ [39,40], Ag_2_O [41,42], SnO_2_ [43,44], ZnO [45], MoO_3_ [46], WO_3_ [47], Cu_2_O [48,49]. Steve Dunn et al., proved that BaTiO_3_/α-Fe_2_O_3_ could form the island-like morphology with the minimum addition of α-Fe_2_O_3_, which allowed photons exciting both BaTiO_3_ and α-Fe_2_O_3_. More importantly, the discrete site of BaTiO_3_ and α-Fe_2_O_3_ could combine with dye molecule to form triple points and permit the dye molecule contact charge carriers directly, as shown in Figure 2a,b, and finally resulted in the observed photo-degradation [39]. The good news is that ferroelectric-based BaTiO_3_ has been applied in PEC bioanalysis. Yu et al. [47] reported that WO_3_ nanoflakes/BTO/Cu_2_O photoelectrode could realize the ultrasensitive detection of prostate specific antigen (PSA) with the limit of detection of 0.036 pg/mL. Due to the polar charge carriers-created electric field of BaTiO_3_, as illustrated in Figure 2d, the electrons and holes of WO_3_ and Cu_2_O can be induced to directional migration (Figure 2e), and finally realized the larger photocurrent density response (Figure 2f).

In addition to oxide semiconductors, metal sulfide semiconductor, such as CdS [50,51,52], Ag_2_S [53] etc., could enhance the photo-electric conversion by integrating with ferroelectric. Liu et al., reported that BaTiO_3_-CdS hybrid photocatalysts obtained H_2_ production rate of 483 µmol∙h^−1^∙g^−1^ with the 20 wt% CdS loading due to the spontaneous polarization electric field of BaTiO_3_ [52]. As illustrated in Figure 2c, with the aid of electric field driving force of BaTiO_3_, photo-induced electrons and holes in CdS would be separated effectively for followed water splitting for hydrogen preparation. Wang et al., introduced (Ag-Ag_2_S)/BaTiO_3_ hybrid ternary structure and reached a high MO degradation rate of 90% in 30 min on account of both synergistic exciton-plasmon interaction in Ag-Ag_2_S and piezoelectric polarization in Ag_2_S/BaTiO_3_ [53].

**Figure 2 nanomaterials-12-03026-f002:**
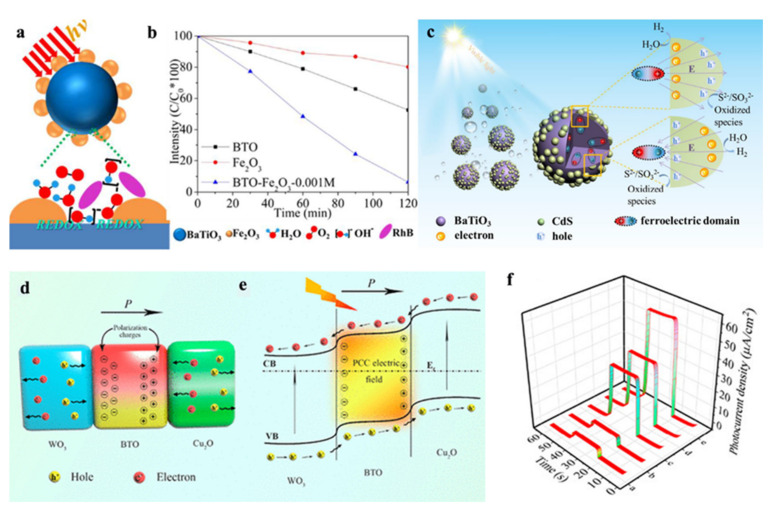
(**a**) Schematic island-like morphology of Fe_2_O_3_/BaTiO_3_ and the triple points between the dye solution, Fe_2_O_3_ and BaTiO_3_ provide active sites for redox reactions; (**b**) photodecolorization profiles of RhB with different photocatalysts under simulated sunlight. Adapted with permission from ref. [39], copyright: 2017, American Chemical Society. (**c**) Schematic of photoinduced hole and electron migration in BaTiO_3_–CdS composites and photocatalytic hydrogen process under visible light; Adapted with permission from ref. [52], copyright: 2018, Elsevier. (**d**) Schematic diagram of the promoting effect of the PCC electric field on the separation of electron-hole pairs; (**e**) DSCC Strategy of the WO_3_ Nanoflakes/BTO/Cu_2_O Photoelectrode; (**f**) photocurrent responses of WO_3_ nanoflakes (curve a), WO_3_/BTO (curve b), WO_3_/Cu_2_O (curve c), WO_3_/Cu_2_O/BTO (**d**), and WO_3_/BTO/Cu_2_O after poling (curve e). Adapted with permission from ref. [47], copyright: 2020, American Chemical Society.

### 2.2. PbTiO_3_

Tetragonal PbTiO_3_ is a well-known ferroelectric material with a band gap larger than 2.75 eV, and possess a certain photocatalytic effect under UV light irradiation with a similar crystal structure to BaTiO_3_. Through ion doping, such as Cu^2+^ [54], Fe^3+^ [55], Mo [56], the light absorption of the material can be enhanced and its photocatalytic effect can be improved. Additionally, due to local surface plasmon resonance phenomenon, noble metals, such as Ag [57] and Pt [58] elements can also be adhered to the surface of PbTiO_3_ to improve the photocatalytic efficiency. In another point, the formation of core-shell structure or heterojunction, such as TiO_2_@PbTiO_3_ [59,60], can significantly increase the photocurrent efficiency (up to 19%), which is almost 20 times higher than literature. By decorating CdS nanoparticle onto the pure Pb(Zr_0.2_Ti_0.8_)O_3_ film, 11 times photocurrent density can be obtained due to effectively separation of photo-induced carriers by spontaneous ferroelectric polarization and the broaden band gap by CdS nanoparticle [61]. Besides the above methods, the crystal morphology has a great influence on the performance of PbTiO_3_. Li et al., prepared the single-domain PbTiO_3_ nanoplates and confirmed the internal depolarization field play an vital role in advancing photogenerated charge separation rather than polarization-caused band bending on the surface [62]. PbTiO_3_ nanoplates [63] can obtained high H_2_ evolution by selectively depositing Pt particles on the positively specific orientation of (110) facets, while consolidating MnO_x_ on the negatively (110) facets under the influence of ferroelectric field due to the directional diffusion of photo-generated electrons and holes.

### 2.3. BiFeO_3_

As a prototypical ferroelectric perovskite oxide, BiFeO_3_ (BFO) stands out as an excellent ferroelectric material for photo-electric conversion due to suitable band gap [64]. And it can be used as a photoelectrode alone, or as an auxiliary material to combine with classical photocatalytic materials such as C_3_N_4_ [65], CdS [66] or it can be modified with other materials (combined with Au [67], doping by Sr^2+^ at Bi-site [68], etc.). Cao et al. [69] prepared high-quality BiFeO_3_ polycrystalline films on ITO with thickness of the ITO and BiFeO_3_ 100 nm and 300 nm, respectively. The barrier height between ITO and BiFeO_3_ is 1.24 eV, which is a typical Schottky junction. As shown in Figure 3a, the ferroelectric polarization P direction points to the electrolyte and its depolarization field –EP (ie Ebi) points to the bottom electrode, which leading to energy band upward on the left edge bends and the right edge bends downward after the BiFeO_3_ film was polarized by an external voltage of +8 V. However, as shown in Figure 3b, when the BiFeO_3_ film is polarized at −8 V, holes are difficult to transport from the electrolyte to BiFeO_3_, while electrons are difficult to transport from BiFeO_3_ to the electrolyte, and obtain the opposite result compared with +8 V polarization. Additionally, the current density also changes with the transition of the ferroelectric polarization states of BiFeO_3_ films. As shown in Figure 3c, on account of Schottky barrier at the BiFeO_3_/ITO interface, the photocurrents of polarized BiFeO_3_ were all negative, regardless of the BiFeO_3_ film was polarized by +8 V or −8 V. But when the working voltage of BiFeO_3_ electrode was 0 V, the photocurrent density of the BiFeO_3_ film after polarization at +8 V was −10 µA/cm^2^. In order to illustrate how the poling operation influence the charge transfer, the authors offered a sleek architectural design in which PEC data was gathered by immersing the polarized electrodes in the electrolyte with 50 µM Rhodamine B as the modifier. As shown in Figure 3d, the +8 V polarized BiFeO_3_ film presented a peak at 590 nm in the external quantum yield spectra, which is corresponding to the absorption spectrum of Rhodamine B. This result photocurrent signal derived from photo-excited hole injection of Rhodamine B rather than the BiFeO_3_ film and in turn had validated the correctness of mechanism of photo-excited charge transfer from BiFeO_3_ film to the KCl electrolyte shown in Figure 3a,b.

Huang et al. [70] achieved the enhancement of photocatalytic current density of 1.76 mA/cm^2^ by assemble BiFeO_3_ on Sn-doped TiO_2_ nanorods (Sn:TiO_2_@BiFeO_3_). The single as-synthesized Sn:TiO_2_@BiFeO_3_ nanorod with thin BiFeO_3_ film can be seen in Figure 4c. The PEC data revealed that coating BiFeO_3_ can significantly improve the photocatalytic performance of TiO_2_ nanorods. As shown in Figure 4a, when the working electrode potential is 1.23 V, the photocurrent density of Sn:TiO_2_@BiFeO_3_ increased to 1.51 mA/cm^2^. Furthermore, when the external voltage polarized BiFeO_3_ film, the ferroelectric polarization points to the electrolyte and its depolarization field points to Sn:TiO_2_, as shown in Figure 4d. Meanwhile, the energy band at the Sn:TiO_2_@BiFeO_3_ interface bends upwards, and its depletion layer widened. Thus, the built-in electric field of BiFeO_3_ increased the electron hole separation efficiency, and the photo-current density increases from 1.51 mA/cm^2^ to 1.76 mA/cm^2^ (Figure 4b). However, as shown in Figure 4e, when the applied polarization voltage points to the FTO, the depletion layer at the Sn:TiO_2_@BiFeO_3_ interface is narrowed, which reduces the electron hole separation efficiency, and the photocatalytic current density of the photoanode decreases from 1.51 mA/cm^2^ to 1.02 mA/cm^2^ (Figure 4b).

Except for normally semiconductor materials, BiFeO_3_ constantly be combined with other type ferroelectrics, such as LaNiO_3_ [71], XTiO_3_ (X = Sr, Zn, Pb) [72], BiVO_3_ [73] et al., which act as the buffer layer used to obtain high-quality BFO thin film to improve the related properties. Han et al. [71] synthesized BFO photocathode by introducing a thin LaNiO_3_ (LNO) film to improve the PEC H_2_O_2_ production. The current–voltage test proved that the photocurrent can raise from −0.3 to −0.9 mA cm^−2^ and the onset potential can positive shift from 1.27 eV to 1.38 eV due to the introduction of the LNO dense layer. At the same time, on account of the improvement of carrier dynamics, the incident photon conversion efficiency at 350 nm reached 12% for LNO/BFO photocathode, which was 2.7 times higher than bare BFO electrode. As a result, the researchers obtained excellent H_2_O_2_ production of 278 µmol/L, with doubled faradic efficiency, due to rich carrier collections and kinetics. Zhao Yu et al. [72] designed BiFeO_3_/XTiO_3_ (X = Sr, Zn, Pb) multilayer films on the LaNiO_3_ substrate and acquired ingenious photodetection of UV-visible light on account of the advantages of broaden UV absorption of XTiO_3_, enhanced electric field of the heterojunction and powerful built-in electric field of BiFeO_3_. The AEM images of BFO/ST film, BFO/ZT film, and BFO/PT film were present on Figure 5a–c. The data of current density-voltage (J–V) in Figure 5d–f showed these films presented favourable photovoltaic performance under monochromatic light. The current density increased with the optical power density up to 8 mW cm^−2^, while the dark current densities of all films kept low values due to the suitable energy band structure. The band structure under illumination of the BFO/ST film, BFO/ZT film, and BFO/PT film were present on Figure 5g–i, and the photoelectrons migrated from CB of XT to BFO, while the holes migrated from VB of BFO to XT. As a joint result of built-in electric field and heterojunction electric field, the photoelectron conversion of BFO/XT multilayer films stable output.

Due to the existence of a lone pair in Bi^3+^, BiFeO_3_ possesses the nature of ferroelectric polarization. Thus, last but not least, substitution of special elements at A-site can inhibit the volatility of Bi [74]. Additionally, the alternative B-site of a transition metal element can diminish the valence state fluctuations of Fe^3+^, and both of above substitutions can decrease the leakage current of BiFeO_3_ [75]. You et al. [76] reported the modify BiFeO_3_ nanofibers by A-site Pr ion and B-site Mn ion co-substitution, and as presented in Figure 6a, under no-poling condition, the photocurrent density was effectively enhanced from 8.2 μA⋅cm^2^, 17.2 μA/cm^2^, 31.9 μA/cm^2^ to 72.5 μA/cm^2^ for BiFeO_3_, BiPrFeO_3_, BiFeMnO_3_ and BiPrFeMnO_3_ photoelectrodes under one sun illumination, respectively. Furthermore, the PEC performance of all these photoelectrodes increased when the samples were negatively poled and the current density of BiPrFeMnO_3_ photoelectrode reached to 131.2 μA/cm^2^. Meanwhile, the linear sweep voltammetry characteristics in Figure 6b indicated the current density for negatively poled BiPrFeMnO_3_ photoelectrode up to 145 μA/cm^2^ and the onset potential left shift from −0.16 V to −0.18 V. As expected, the photocatalytic capability on RhB dye of the Pr and Mn co-doping BiFeO_3_ nanofibers boosted the degradation performance. As shown in Figure 6c,d, the degradation ratios of BiFeO_3_, BiPrFeO_3_, BiFeMnO_3_ of RhB dye were 17%, 25% and 29%, respectively. However, the obtained co-doped sample could degrade 49% for dye and the photocatalytic rate given rise to 0.0256 min^−1^. The excellent performance of co-doped BiFeO_3_ illustrates that proper ion-doping was an efficient way to improve the PEC property for ferroelectrics.

### 2.4. Bi_2_FeCrO_6_

Bi_2_FeCrO_6_ (BFO) has a special double perovskite structure, which has great advantages in visible light driven redox reactions due to its relatively small band gap. The Eg is controlled by mutual effect between Fe and Cr via O, while the ferroelectricity is actuated by Bi^3+^. In 2014, R. Nechache et al. [77] engineered the BFO thin-film by multilayer configuration for power conversion efficiency of 8.1% solar cells, which breached the efficiency previously reported. This article described in detail the depositional condition impact on the property of long-range ordering by characterizing the intensity ratio R and the ordered domain size D, as presented in Figure 7a. Due to the lower direct bandgap and low photogenerated carrier recombination rate, the photoelectric conversion efficiency reached to 3.3% for device with the active layer of low temperature of 580 °C and low-growth-rate of 2 Hz, while choosing SrRuO_3_ film as bottom electrodes and ITO arrays as transparent conducting electrodes and the schematic diagram was shown in Figure 7b,c. Moreover, after positively polarization of 25 V, the device with multi-active layer of laser repetition rate obtained excellent PEC efficiency of 8.1% as shown in Figure 7d,e. These excellent results demonstrated the enormous application potential in photovoltaic device of ferroelectric-based materials as active layers. Afterwards, in 2017, R. Nechache et al. [78] assembled special p-i-n structure with i-BFO thin-film as sandwich between p-type NiO and n-type Nb-doped SrTiO_3_(NSTO), as shown in Figure 7f, and yielded the PEC value in Figure 7g of 2.0%, which was four times enhancement compared with the double layer of i-BFO thin-film and n-type Nb-doped SrTiO_3_ on account of the efficient charge separation driven by the internal polarization as well as above-bandgap generated photovoltage. In addition, in 2019, as exhibited in Figure 7h,i, this team fabricated stable p-NiO/n-BFO heterojunction photoanodes [79] and achieved a high incident photon-to-current efficiency of 3.7% with an enhancement of photocurrent density of 0.4 mA cm^−2^ compared with the bare BFO devices. The photocatalytic degradation and PEC efficiency of ferroelectric materials has been shown in Table 1.

### 2.5. 2D Ferroelectrics

Since the discovery of graphene [83], the expanded family of two-dimensional materials has shown abundant physical properties. In terms of the exploration of 2D ferroelectric properties, due to the saturated interfacial chemical environment of layered materials and the weak interaction between layers, it is possible to prepare stable low-dimensional ultrathin ferroelectric films. Recently, a number of representative research reports on ferroelectricity in two-dimensional materials have emerged, and considering the lattice symmetry required for ferroelectrics, some experimental [84] and theoretical calculations [85] have confirmed the existence of ferroelectricity in 2D layered van der Waals materials. For readers seeking more information on the development of 2D layer ferroelectric materials, we recommend them to refer to other review for a broad scope in this area [86]. In 2015, A. Belianinov et al. [87] reported that the layered material CuInP_2_S_6_ exhibited ferroelectricity at room temperature. The researchers measured spontaneous ferroelectric domains with obvious hysteresis loops in samples thicker than 100 nm by piezoresponse force microscopy (PFM) and the ferroelectric polarization direction can be reversed under an applied electric field, while the ferroelectricity disappears with a thickness less than 50 nm due to the potential depolarization field. This work confirmed the possibility of ferroelectricity in two-dimensional layered materials at room-temperature and stimulated the exploration of thin-layer and even single-layer limit-thickness ferroelectrics. Then in 2016, also in the CuInP_2_S_6_ system, Liu et al. [84] re-studied its ferroelectric properties and reported that double-layer with the thickness of 2 nm CuInP_2_S_6_ undergoes out-of-plane spontaneous electrical polarization at room temperature (Figure 8a,b). The dominant peaks in Raman spectrum illustrated the crystal symmetry of CuInP_2_S_6_. In this work, they used PFM to probe the piezoelectric response, the presence of stable ferroelectric spontaneous poles can be determined from the amplitude map (Figure 8c), PFM phase (Figure 8d) change and the obvious butterfly-shaped hysteresis loop in Figure 8e. Furthermore, to verify the polarization-controlled erasing and writing of ferroelectrics, they demonstrated the construction of a back-shaped ferroelectric polarization domain could still be maintained, as shown in Figure 8f. In addition, through second harmonic wave (SHG) measurements, they demonstrated that the ferroelectric polarization inversion was accompanied by a structural phase transition, confirming that the transition temperature of the ferroelectric-paraelectric phase transition in 2D CuInP_2_S_6_ was above room temperature (~320 K). Additionally, Wang Dong et al. [88] broadened the family of metal phosphorous trichalcogenides M_1_M_2_P_2_X_6_ (M1 = Cu/Ag, M2 = In/Bi and X = S/Se) and proved that the M1-X bond in the Cu-based and S-based systems have the responsibility of high phase transition temperatures. Furthermore, the composite structure in CuInP_2_S_6_/Mn_2_P_2_S_6_ and CuInP_2_S_6_/Zn_2_P_2_Se_6_ could be detected via fast charge separation, which could provide the possibility in the area of photocatalytic water splitting. Kou et al. [89] also proved that photocatalytic activities and photoelectron conversion efficiency can be enhanced by monolayer AgBiP_2_Se_6_ system based on first principles calculations.

Although few-layer CuInP_2_S_6_ exhibits two-dimensional out-of-plane ferroelectricity, it’s not conducive to high-quality preparation and application promotion due to it is polycompound. In view of these shortcomings and also develop more practical two-dimensional out-of-plane ferroelectricity, in 2017, Ding et al. [85] predicted the room-temperature spontaneous ferroelectric polarization with in-plane and out-of-plane coupling in a binary monolayer α-In_2_Se_3_ and predicted that this characteristic could be generalized to other III_2_-VI_3_ 2D van der Waals layered materials. At the same time, this study also demonstrated the controllability of ferroelectric α-In_2_Se_3_ by constructing a double-layer heterojunction composed of 2D ferroelectric α-In_2_Se_3_ and other 2D materials. Inspired by the above theoretical work, Zhu et al. [90] fabricated α-In_2_Se_3_ with a thickness of only 1.1 nm on SiO_2_/Si substrate by mechanical lift-off method, as shown in Figure 9a,b, and verified the room temperature ferroelectricity in α-phase In_2_Se_3_ experimentally, which is close to the reported monolayer c-In_2_Se_3_ thickness (~1 nm). The slightly red shift of Raman spectrum indicated the α-phase crystal structure of the In_2_Se_3_ thin layers. Due to the piezoelectric properties, ferroelectrics will undergo corresponding deformation in an external electric field. The authors observed spontaneous out-of-plane ferroelectric polarization on α-In_2_Se_3_ with a thickness of about 20 nm by PFM in the atmospheric environment, and a phase bias of 180° was be obtained between different ferroelectric domains (Figure 9c), indicating the existence of anti-parallel arrangement of electric dipoles in the out-of-plane direction. By applying an external electric field, the out-of-plane amplitudes (Figure 9e) have obvious butterfly-shaped and well phase nonlinear ferroelectric hysteresis (Figure 9f), indicating that the electric polarization direction was reversed under the external electric field. At the same time, the coercive field (Ec) of ferroelectric α-In_2_Se_3_ was about 200 kV/cm, much lower than the coercive field of the 2D layered material CuInP_2_S_6_ reported earlier (700 kV/cm) [84], which means that ferroelectric devices based on α-In_2_Se_3_ will consume less power. In order to verify the ferroelectric retention of α-In_2_Se_3_, as shown in Figure 9d, the authors controllably wrote the out-of-plane polarization by applying a positive bias voltage on a microscopic area of 1 μm × 1 μm, and then applied a reverse bias voltage on the central 0.5 μm × 0.5 μm area to write the electrode by PFM. In addition, the ferroelectric domain still remained after 24 h due to the ferroelectric polarization.

## 3. Conclusions

Ferroelectric-based photocatalysis and photovoltaic serves a crucial approach for effective and serviceable determination of a target object, such as H_2_O, CO_2_, contaminants and so on. This review summarized the progress of ferroelectric materials by introducing the photovoltaic conversion properties of perovskite ferroelectric materials such as BaTiO_3_, BiFeO_3_, Bi_2_FeCrO_6_, CuInP_2_S_6_ and α-In_2_Se_3_. Even more to the point, single components exhibit low efficiency because of a broadened band gap and low electron conductivity. Strategies to improve photoelectric conversion performance could be depicted as follows: one proposed solution is ion doping, which can effectively increase the lifetime of carriers, reduce the recombination rate of electron hole pairs and increase the degree of separation of electron hole pairs inside the semiconductor. Another alternative is combining two semiconductors with apposite energy level and bandgap to change band bending, widening the light absorption range and accelerating carriers separation.

On account of the continued development in ferroelectrics as well as photocatalysis and photovoltaic reactions, research in this area is still growing at a high rate, meanwhile, more problems still need to be solved. We believe the future work will focus on improvement of charge separation efficiency, energy conversion efficiency, stability of structure and performance and environmentally friendly. In order to realize the expected above property, several aspects can be prioritized of electrodes in the future: (1) Design and preparation of ferroelectric photocatalytic materials. In consideration of the wider band gaps and lower carrier mobility of currently ferroelectric materials, it is necessary to prepare nanosheets or other nanostructures of ferroelectric with large specific surface area, and make the ferroelectric polarization face away from or face its surface and interface, so as to ensure that the depolarization field P can efficiently separate photogenerated electron hole pairs. (2) Systematic study of PEC processes. The performance determinants of ferroelectric material photoelectrodes should be carefully studied to elucidate the contribution of ferroelectric effects to separated electron hole pairs for further improvements for separation and transfer efficiency of photo-generated electron hole pairs. In more detail, the research on the interface of optoelectronics pole/electrolyte is very important. (3) Study on the stability of catalytic performance of ferroelectric. In practice, the requirements of ferroelectrics not only include the initial photocatalytic performance, but also the stability of catalysts. Up till now, most research has focused on improving the photocatalytic water splitting or pollutants of ferroelectric materials, but fewer articles have been done on stability. The advancement of these aspects would certainly lead to significant advantages compared to the current systems in terms of simplicity, high efficiency and stabilization. We believe that ferroelectric materials will play a pivotal role in the field of environmental applications in the near future.

## Figures and Tables

**Figure 1 nanomaterials-12-03026-f001:**
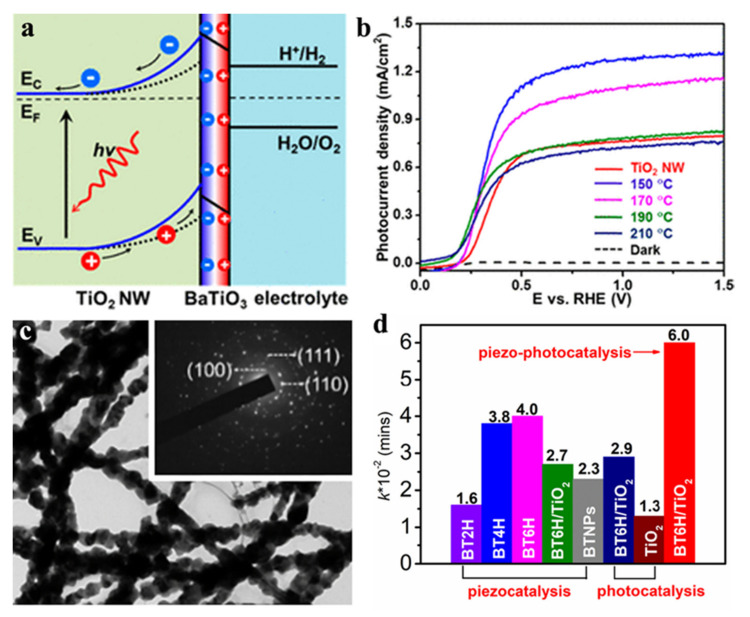
(**a**) The scheme of ferroelectric polarization-endowed band engineering of TiO_2_/BTO core/shell nanowires. (**b**) Photocurrent density-potential curves of TiO_2_ NWs and TiO_2_/BTO NWs under AM1.5G illumination. Adapted with permission from ref. [37]. Copyright: 2015, American Chemical Society. (**c**) TEM and SAED of necklace-like BaTiO_3_ nanofibers; (**d**) the comparison of the corresponding K values for degradation of MO by BT6H@TiO_2_ core-shell nanofibers under various catalytic conditions. Adapted with permission from ref. [38]. Copyright: 2022, Elsevier.

**Figure 3 nanomaterials-12-03026-f003:**
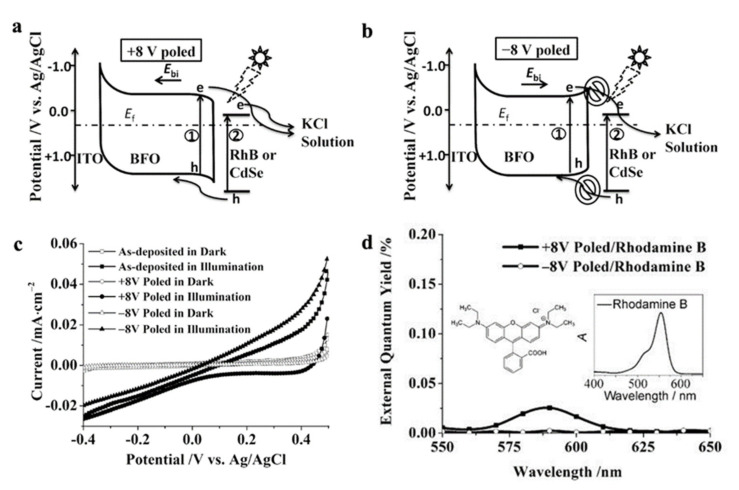
Schematic representations of the mechanisms in photo-excited charge transfer from BFO films to the electrolyte ① and from excited surface modifiers to the BFO films ② after the BFO films were (**a**) positively and (**b**) negatively poled of 8 V. (**c**) Photocurrent–potential characteristics of the photoelectrodes with different polarization states. (**d**) External quantum yield spectra of the BFO electrodes measured with 50 mm Rhodamine B. Inset: the absorption spectrum of Rhodamine B in water. Adapted with permission from ref. [69], copyright: 2014, Wiley.

**Figure 4 nanomaterials-12-03026-f004:**
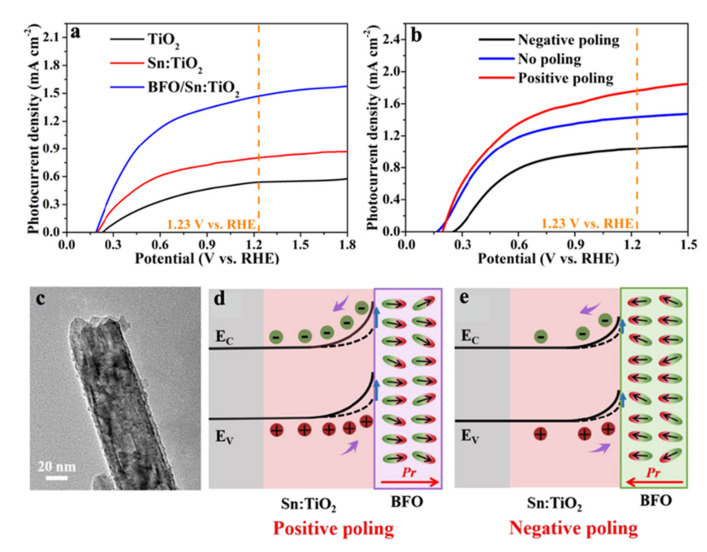
(**a**) LSV performance of TiO_2_, Sn–doped TiO_2_ and BFO/Sn:TiO_2_ NRs; (**b**) J−V curves of BFO/Sn:TiO_2_ NRs with no poling, positive poling and negative poling; (**c**) TEM image of a single BFO/Sn:TiO_2_ NRs; Schematic electronic band diagram of BFO/Sn:TiO_2_ NRs with (**d**) positive poling, and (**e**) negative poling. Adapted with permission from ref. [70], copyright: 2019, Elsevier.

**Figure 5 nanomaterials-12-03026-f005:**
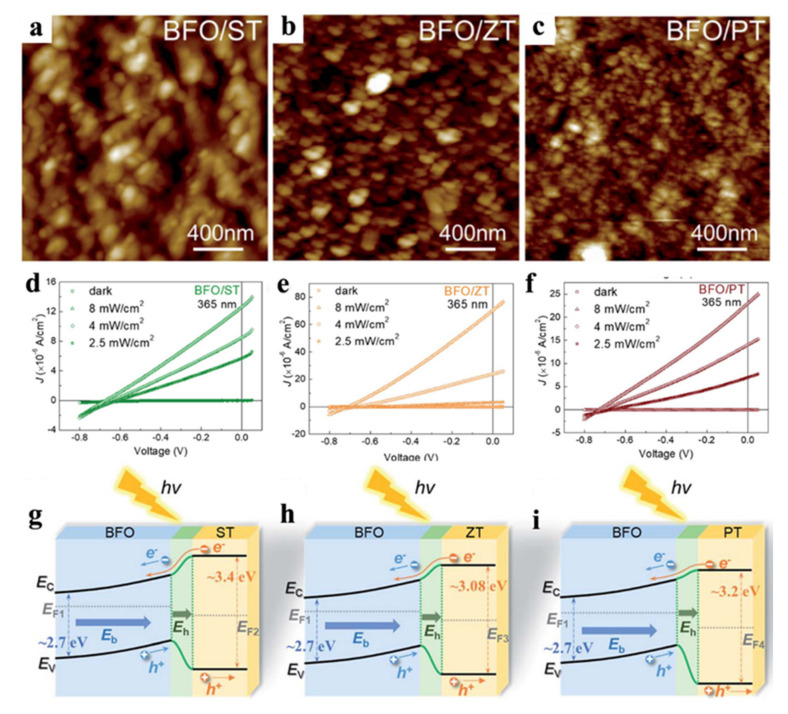
(**a**–**c**) AFM images of the BFO/XT film; (**d**–**f**) J–V curves under the illumination of 365 nm with different power density of BFO/XT film; (**g**–**i**) The band structure of the BFO/XT multilayer films under illumination. Adapted with permission from ref. [72], copyright: 2022, Royal Society of Chemistry.

**Figure 6 nanomaterials-12-03026-f006:**
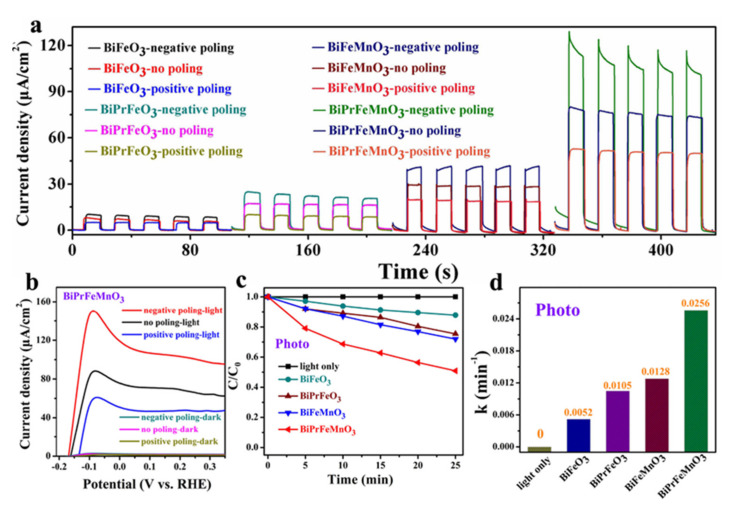
(**a**) Transient photocurrent responses of BiFeO_3_, BiPrFeO_3_, BiFeMnO_3_ and BiPrFeMnO_3_ photoelectrode with different polarization conditions; (**b**) linear sweep voltammetry curves of BiPrFeMnO_3_ photo-anodes with different polarization condition; (**c**) photocatalysis of BiFeO_3_, BiPrFeO_3_, BiFeMnO_3_, BiPrFeMnO_3_ nanofibers and control sample (blank test without catalyst) in the reaction of aqueous RhB within 25 min, respectively; (**d**) the corresponding reaction rate constant of photocatalytic activity. Adapted with permission from ref. [76], copyright: 2022, Elsevier.

**Figure 7 nanomaterials-12-03026-f007:**
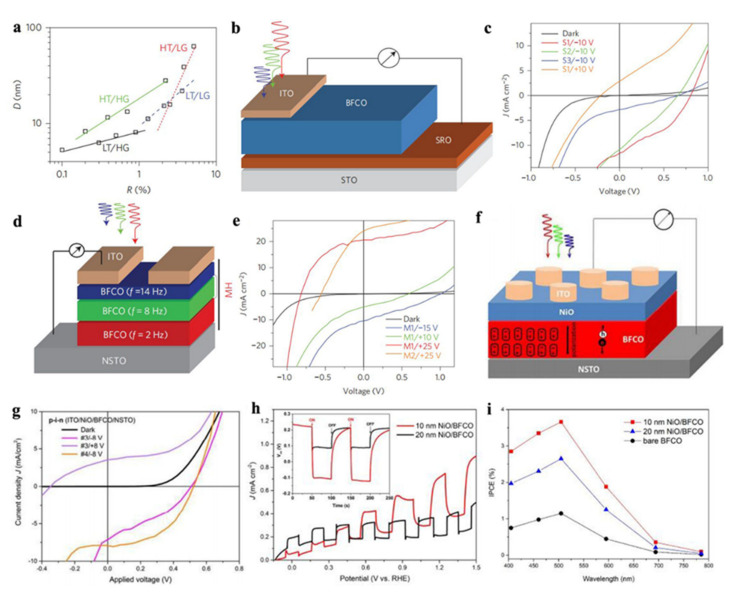
(**a**) R–D relationship in BFCO films grown under different PLD conditions; (**b**) device layout of the tested BFCO single-layer-based structure; (**c**) J–V characteristics of BFCO single-layer devices; (**d**) Device geometry of the tested BFCO multilayer structure; (**e**) J–V characteristics of BFCO multilayer devices. Adapted with permission from ref. [77], copyright: 2015, Nature Phonos; (**f**) Layout of the devices for p–i–n; (**g**) J–V characteristics for p–i–n devices; Adapted with permission from ref. [78], copyright: 2017, Royal Society of Chemistry. (**h**) J–V vs. RHE for the photoanodes coated with 10 and 20 nm NiO layer; The inset shows the corresponding Voc vs. time; (**i**) IPCE spectra for BFCO photoanodes with/without NiO layer at 1.23 V (vs. RHE). Adapted with permission from ref. [79], copyright: 2019, American Chemical Society.

**Figure 8 nanomaterials-12-03026-f008:**
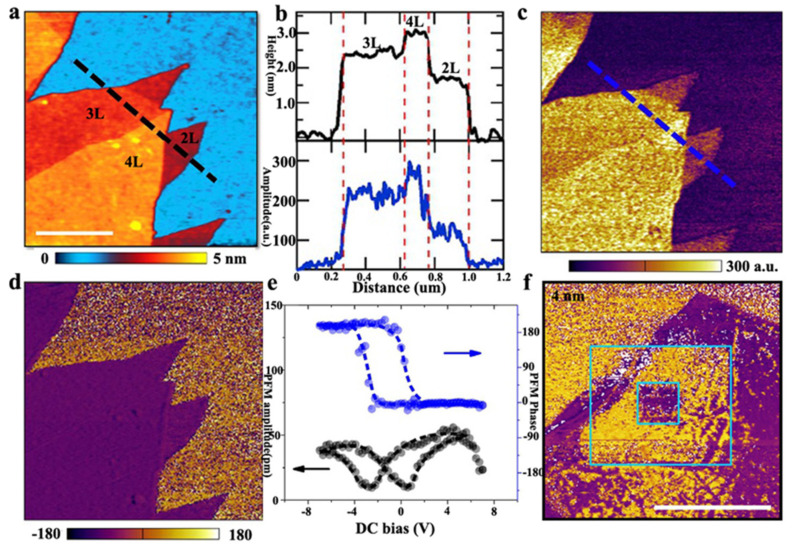
AFM topography (**a**) PFM amplitude (**c**) and phase (**d**) of 2–4 layer thick CIPS on Au coated SiO_2_/Si substrate. Scale bar in a, 500 nm. (**b**) the height (black) and PFM amplitude (blue) profile along the lines shown in a and c, respectively. L, layers. (**e**) The PFM amplitude (black) and phase (blue) hysteresis loops during the switching process for 4 nm thick CIPS flakes. (**f**) The PFM phase images for 4 nm thick CIPS flakes with written box-in-box patterns with reverse DC bias. Scale bar, 1 μm. Adapted with permission from ref. [84], copyright: 2017, Nature.

**Figure 9 nanomaterials-12-03026-f009:**
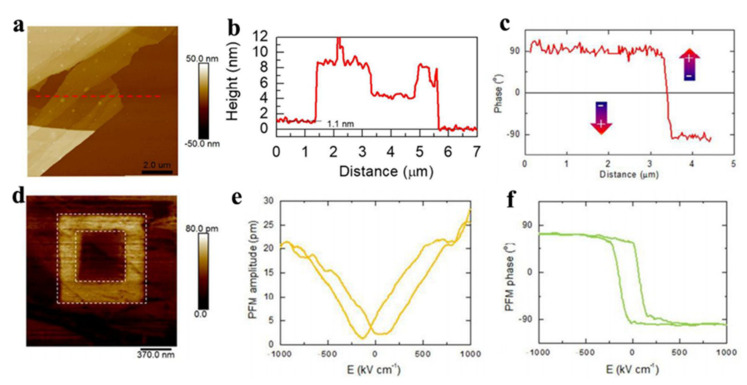
(**a**) AFM topography (10 × 10 µm^2^) of exfoliated α–In_2_Se_3_ thin layers on SiO_2_/Si substrate; (**b**) corresponding height profiles taken along the red dash line in (**a**); (**c**) the phase profile of different ferroelectric domains; (**d**) PFM amplitude image of domain engineering in α–In_2_Se_3_ with film thickness 12 nm; (**e**) PFM phase hysteresis loop measured and (**f**) PFM amplitude from α–In_2_Se_3_ thin layers. Adapted with permission from ref. [90], copyright: 2018, Royal Society of Chemistry.

**Table 1 nanomaterials-12-03026-t001:** Photocatalytic degradation and PEC performance using a variety of catalytic methods of ferroelectric material under one sun illumination [38,39,42,61,64,65,67,70,76,80,81,82].

Material and Structure	Bandgap/eV	Light Source	PCE/%	Photocurrent Density/mA·cm^−2^	Catalytic Degradants	Catalytic Activity	Reference
BiFeO_3_ nanoparticle	2.20	visible light	−	0.4	−	−	[64]
Au/BiFeO_3_	2.62	visible light	10	0.55	−	−	[67]
g–C_3_N_4_/BiFeO_3_	2.16	visible light	−	−	Rhodamine B	90%/60 min	[65]
Ag/BiFeO_3_	2.46–2.32	full spectrum	−	0.35	−	−	[82]
KNbO_3_ nanowire	3.28	full spectrum	−	0.115	−	−	[81]
BaTiO_3_/α–Fe_2_O_3_	3.14	full spectrum	−	−	Rhodamine B	0.0153 min^−1^	[39]
BaTiO_3_@TiO_2_ core-shell	−	full spectrum	−	−	Methyl orange	0.06 min^−1^	[38]
PbTiO_3_/CdS	3.2	full spectrum	7.4	0.106	−	−	[61]
Ag/Nb:SrTiO_3_	−	full spectrum	−	1.30	−	−	[80]
BiPrFeMnO_3_	2.08	full spectrum	−	1.31	Rhodamine B	0.1352 min^−1^	[76]
Sn:TiO_2_@BiFeO_3_	2.84	full spectrum	IPCE:82%	1.76	−	−	[70]
TiO_2_/BaTiO_3_/Ag_2_O	−	full spectrum	29	1.8	−	−	[42]

## Data Availability

Not applicable.

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
