# Peer review of "Recent Advances in Ferroelectric Materials-Based Photoelectrochemical Reaction"

_nanomaterials, 2022, doi:10.3390/nano12173026_

Round 1
Reviewer 1 Report
This review should deal with “Ferroelectric-based photoelectrochemical materials highlighted with recent examples”. Actually, at the end of the Introduction it is stated that” this article is not designed to cover comprehensive theoretical knowledge of ferroelectric-based photovoltaic and photocatalysis but rather to offer significant ferroelectric materials”. I suppose that, in agreement with its title, this article is designed to show/list/describe ferroelectric materials that are also significant for photoelectrochemical properties and applications.
The topic is surely interesting as well as many of the examples described in the paper.
However, in my opinion the paper has to be improved to be suitable for publication. In particular, the English has to be deeply revised, because many mistakes are present along the whole text and some phrases are not understandable.
In addition, in the section dedicated to 2D ferroelectrics the authors don’t describe their PEC properties at all.
Please, find below few examples (NOT an exaustive list) of unclear phrases
Lines 25-30: the authors refer first to ABO3 that is the general formula for perovskite oxides. Then, they examine “…other ferroelectric ceramics, such as KNbO3, KTaO3, PbTiO3, etc.” All the cited compounds are described by ABO3 formula, with A=K or Pb and B=Ta, Nb or Ti.
Line 32: please select one of the two adjectives: splendid or excellent since they are synonyms.
Line 34: band gap are not economic
Line 45: recombination or combination?
Line 58: what does it mean that the paper is designed to “offer significant ferroelectric materials.”?
Line 67: I suggest the authors to write “center of the cube” instead of “body centered” to avoid confusion with body centered “I” Bravais lattice
Line 97: “except” or “besides”?
Line 323 and below: it seems that PFM is never defined
Line 358: “complex quaternary compound lattice structure” the phrase is not clear.
Reviewer 2 Report
The paper presented to me for review "Ferroelectric-based photoelectrochemical materials highlighted with recent examples" by Li-min Yu et al., which is a review of research on inorganic perovskite ferroelecric based nanomaterials used in photoelectrochemical processes, is written carefully. It contains numerous references. The presented Figures are with the consent of the publishers of individual papers.
I recommend publishing the manuscript in the present form.
Reviewer 3 Report
Authors selected an appropriate topic for their review. However some concerns remain to further its quality:
1- please consider changing the title of the review, here is one suggestion:
"Recent advances in ferroelectric materials based photoelectrochemical reaction"
2- The review needs to be enriched by Tables to bring to the reader attention, the most important findings, hence, please provide for each compound (BTO, PbTO, BFO…) its specific table as much as possible and provide with a final table comparing all these compounds. Please include references in the tables.
3- Page 8, line 250: please provide a short explanation on how the substitution can decrease the leakage current of BFO
4- Please respect the color codes in all figures
5- For the 2D ferroelectrics section: authors are encouraged to add more characterization of the cited compounds especially Raman spectra for pristine and doped ferroelectrics.
6- In the conclusion, for the statement in page 13 line 411 to 414: I agree with the authors, however there are some works already performed on downsizing BFO (doi.org/10.3390/catal12020215) to improve for instance its photocatalytic performance that authors could add to their review.
7- There are several english errors, below some examples but authors are encourage to visit the whole manuscript and make the appropriate corrections:
- Page 1 line 10 & 11: “…are supposed to play a vital role… change to “…are intended to play a crucial role…”
- Page 1 line 22: “deal” change to “cope”; “problem” change to “concerns”
- Page 1 line 27: “possess ferroelectric polarization” change to “possess intrinsic ferroelectric polarization”
- Page 1 line 30 & 31: “The ferroelectric polarization originates the asymmetry” change to “The ferroelectricity is due to the asymmetry of …”
Round 2
Reviewer 1 Report
Some effort was made to improve the article language and style, but I still believe that the English has to be revised, possibly by an English mother tongue.
Besides, I would like to highlight a couple of points:
It is very hard to read Table 1
Lines: 361-363 Do the authors mean that it is not possible to prepare single phase CuInP2S6 samples?
Reviewer 3 Report
Authors have addressed all concerns, I recommend to publish the review in its present form
